# Design, Synthesis, and Characterization of a Novel Blue-Green Long Afterglow BaYAl_3_O_7_:Eu^2^^+^, Nd^3^^+^ Phosphor and Its Anti-Counterfeiting Application

**DOI:** 10.3390/nano13172457

**Published:** 2023-08-30

**Authors:** Jiao Wu, Quanxiao Liu, Peng Gao, Jigang Wang, Yuansheng Qi, Zhenjun Li, Junming Li, Tao Jiang

**Affiliations:** 1Beijing Key Laboratory of Printing and Packaging Materials and Technology, Beijing Institute of Graphic Communication, Beijing 102600, China; wujiao20220912@163.com (J.W.); drllqx@163.com (Q.L.); gaopeng08252022@163.com (P.G.); 2National Center for Nanoscience and Technology, CAS Key Laboratory of Nanophotonic Materials and Devices (Preparatory), Beijing 100190, China; 3The GBA Research Innovation Institute for Nanotechnology, Guangzhou 510700, China; 4Beijing Key Laboratory for Sensors, Beijing Information Science & Technology University, Beijing 100192, China; li@bistu.edu.cn; 5CAS Center for Excellence in Nanoscience, Beijing Key Laboratory of Micro-Nano Energy and Sensor, Beijing Institute of Nanoenergy and Nanosystems, Chinese Academy of Sciences, Beijing 101400, China; jiangtao@binn.cas.cn

**Keywords:** long afterglow, BaYAl_3_O_7_:Eu^2^+, Nd^3^+, screen printing, anti-counterfeiting

## Abstract

Herein, a series of novel long afterglow nanophosphors BaYAl_3_O_7_:Eu^2^^+^, Nd^3^^+^ was synthesized by the combustion method. The investigation encompassed the characterization of X-ray diffraction, morphology, chemical valence, elemental composition, and photoluminescence behavior of BaYAl_3_O_7_:Eu^2^+ and BaYAl_3_O_7_:Eu^2^^+^, Nd^3^^+^ nanoparticles. Under 365 nm excitation, BaYAl_3_O_7_:Eu^2^^+^ and BaYAl_3_O_7_:Eu^2^^+^, Nd^3^^+^ show emission bands centered at 497 nm and 492 nm, which are attributed to the 4f^6^5d→4f^7^ transition of Eu^2^^+^ ions. The optimal samples of BaYAl_3_O_7_:0.03Eu^2^^+^ and BaYAl_3_O_7_:0.03Eu^2^^+^, 0.02Nd^3^^+^ have average fluorescence lifetimes of 850 ns and 1149 ns, respectively. The co-doping of Nd^3^+ ions as the trap centers produced long afterglow luminescence properties, and the afterglow time could reach up to 8 min. Furthermore, the fluorescent powder can be mixed with polyacrylic acid to prepare anti-counterfeiting inks; a clover pattern and snowflake pattern have been successfully printed using screen printing technology, proving its potential application in the field of anti-counterfeiting.

## 1. Introduction

Long afterglow luminescent materials are a class of materials with special luminescent properties. The phosphors can absorb light energy for a short period and continue to emit light for a few seconds to a few days even after stopping external excitation [1,2]. First, because of their unique environmental protection, energy saving, and other characteristics, application potential of the materials is considered in lighting, security signs, night display, anti-counterfeiting, and other fields [3,4,5,6,7,8]. Second, long afterglow luminescent materials usually have tunable luminescent colors, which can be adjusted by adjusting the material composition, crystal structure, and doping ions to reach diverse luminescent effects. In addition, these materials usually have good chemical stability and optical properties and are suitable for use under various environmental conditions. Since 1996, when the first report appeared about the strong long afterglow luminescent materials SrAl_2_O_4_:Eu^2^^+^, Dy^3^^+^, long afterglow rare-earth luminescent materials have been widely investigated. In SrAl_2_O_4_:Eu^2^^+^, Dy^3^^+^ phosphor, Eu^2^^+^ serve as the luminescence center, while Dy^3^^+^ ions are thought to strongly enhance the sustained luminescence intensity [9,10].

Long afterglow luminescent materials often have two sorts of centers: emission centers and trap centers [11]. When stimulated by external energy, these emitting centers operate as photoactive centers, radiating in the visual range. Trap centers, on the other hand, which are accompanied by lattice defects, store energy and gradually release it to the emission centers once excitation has stopped. The color of the light emitted is usually determined by the emission center. As we all know, Eu^2^+ ions as efficient activators are common ions widely studied in the field of long afterglow materials [12,13,14,15]. In addition, to obtain or improve the long afterglow performance, one of the most common methods is by co-doping with RE^3^+ ions. It produces new traps or changes the inherent trap properties by non-equivalent substitution. In recent years, several remarkable long afterglow phosphors were developed using this strategy. For example, CaAl_2_O_4_:Eu^2^+, Nd^3^+ (blue), Sr_2_MgSi_2_O_7_:Eu^2^^+^, Dy^3^^+^ (blue), SrAl_2_O_4_:Eu^2^^+^, Dy^3^^+^ (green), Ca_2_BO_3_Cl:Eu^2^^+^, Dy^3^^+^ (yellow), Ca_3_Si_2_O_7_:Eu^2^^+^, Sm^3^^+^/Tm^3^^+^ (orange), CaS:Eu^2^^+^, Tm^3^^+^ (red), Sr_5_(PO_4_)_3_Cl:Eu^2^^+^, Nd^3^^+^ (NIR) [9,16,17,18,19,20,21], etc. Furthermore, various synthetic processes, including sol-gel, combustion, and co-precipitation, have been used to synthesize these long afterglow luminescent materials [22,23,24,25]. Among these, the combustion method is simple to use, appropriate for high-volume and low-cost preparation, and permits the synthesis of nanoparticles with controlled shape and particle size at lower temperatures [11,26].

In this study, a novel BaYAl_3_O_7_:Eu^2^+, Nd^3^+ long afterglow phosphor was prepared by the combustion method. Its photoluminescence (PL), optical band gap (*E*_g_), thermoluminescence (TL), long afterglow properties, etc., were investigated. Then, the anti-counterfeiting inks were prepared, showing their potential applications in the field of anti-counterfeiting.

## 2. Experimental Section

### 2.1. Materials and Method

A series of Ba_0.5−0.5x−0.5y_Y_0.5−0.5x−0.5y_Al_3_O_7_: xEu^2^^+^, yNd^3^^+^ (x = 0, 0.02, 0.03, 0.04, 0.05, 0.06; y = 0.01, 0.02, 0.03, 0.04, 0.05) samples was prepared by the combustion method. The raw materials used were BaCO_3_ (purity: 99.99%), Y(NO_3_)_3_ (99.99%), Al_2_O_3_ (99.99%), Eu_2_O_3_ (99.99%), Nd_2_O_3_ (99.99%), HNO_3_ (80%), and urea (99.99%), which were all purchased from Tianjin Chemical Reagent Factory. All reagents were directly used as received without further purification.

### 2.2. Synthesis of Nanomaterials

First, appropriate amounts of HNO_3_ and deionized water were added to BaCO_3_, Y(NO_3_)_3_, Al_2_O_3_, Eu_2_O_3_, and Nd_2_O_3_ to prepare 0.2 mmol/mL of Ba(NO_3_)_2_, 0.5 mmol/mL of Y(NO_3_)_3_, 1 mmol/mL of Al(NO_3_)_3_, 0.1 mmol/mL of Eu(NO_3_)_3_, and 0.5 mmol/mL of Nd(NO_3_)_3_ solutions, respectively. Then, Ba(NO_3_)_2_, Y(NO_3_)_3_, Al(NO_3_)_3_, Eu(NO_3_)_3_, and Nd(NO_3_)_3_ solutions were measured in stoichiometric ratios and placed in a crucible, then 2.2 g of urea for the combustion agent was added and mixed homogeneously, and placed in a muffle furnace preheated to 600 ℃. After waiting for 3–5 min, a loose white and porous solid powder was obtained. Ultimately, the obtained samples were naturally cooled to room temperature and ground into powder for a test.

### 2.3. Preparation of Anti-Counterfeit Ink

The fluorescent powder was introduced into the mixture of ethanol and polyacrylic acid, where the volume ratio of fluorescent powder to the mixed solution was 3:1, followed by a long time of stirring. The amount of polyacrylic acid and ethanol is adjusted to achieve a suitable viscosity for screen printing to obtain the final fluorescent ink. Figure 1 provides a schematic diagram of the preparation of phosphors and screen printing.

### 2.4. Characterizations

The crystalline phases were determined by X-ray diffraction (D/Max-2400 by Rigaku, Tokyo, Japan) analysis with Cu Kα radiation (λ = 1.54 Å). The binding energy was obtained with x-ray photoelectron spectroscopy (XPS, Thermo Fisher Scientific ESCALAB 250 XI, Waltham, MA, USA). The photoluminescence excitation (PLE) and PL spectra were measured by a fluorescence spectrometer (FluoroMax-4, F4700, Hitachi, Tokyo, Japan) equipped with a Xe lamp. The morphology of phosphors was observed through scanning electron microscopy (Quanta 250 FEG, Hitachi, Tokyo, Japan). The sample’s luminescence was measured with a UV lamp supplied by HUNNENGSHI. The fluorescence lifetime and afterglow decay curves of phosphors were measured by a transient steady-state fluorescence spectrometer (FLS1000, Edinburgh Instruments, Edinburgh, UK). The UV-Vis absorption spectra were obtained using a UV-3600 UV-Vis Near Infrared Spectrophotometer (Shimadzu Corporation, Kyoto, Japan). The energy-dispersive X-ray spectroscopy (EDS, SU8020, Hitachi, Tokyo, Japan) spectrum was obtained with a desktop scanning electron microscope energy dispersive spectrometer (Phenom Pro X, Thermo Fisher Scientific, Landsmeer, The Netherlands). The TL was tested with a thermal spectrometer (FJ-427A, China National Nuclear Corporation, Tianjin, China) at a heating rate of 1 °C/s.

## 3. Results and Discussion

### 3.1. XRD Patterns and Morphology Analysis

The X-ray diffraction (XRD) patterns of BaYAl_3_O_7_:xEu^2^^+^(x = 0, 0.02, 0.03, 0.04, 0.05, 0.08) nanophosphors obtained by calcining at 600 ℃ are presented in Figure 2a. The information regarding BaYAl_3_O_7_ (abbreviated as BYAO) is not found in the present literature. Figure 2a exhibits the XRD patterns of the BYAO hosts, doped BYAO samples, and standard cards of the BaAl_2_O_4_ (PDF#00-017-0306) and Y_3_Al_5_O_12_ (PDF#01-073-1370) [27,28]. It is observed that the sample has a mixture phase structure and the diffraction peaks match well with the standard cards of the BaAl_2_O_4_ and Y_3_Al_5_O_12_. Generally, it is reasonable to assume that the Eu^2^^+^ dopants tend to occupy the Ba^2^^+^/Y^3^^+^ sites based on similar effective ionic radii (IR) of the cation with varying coordination numbers (CNs) [29,30,31]. Moreover, the main diffraction peak shifts toward a higher angle with the introduction of Eu ions, as shown in Figure 2a. It implies that the replaced sample lattice characteristics and cell volume are lowered to some amount, which is consistent with the shift of diffraction peaks, which might be well-accepted based on Bragg’s law (2*d* sin θ = *n*λ) [32,33]. It verifies once more that Eu ions with lower ionic radii are continually replacing potentially substitutable ions. The XRD patterns of the BaYAl_3_O_7_:0.03Eu^2^^+^, xNd^3^^+^ (x = 0.01, 0.02, 0.03, 0.04, 0.05) nanophosphors obtained by the calcination at 600 ℃ are presented in Figure 2b. Compared with Figure 2a, the positions of the diffraction peaks remain basically the same, which indicates that the co-doping of Nd^3^^+^ ions does not change the crystal phase of the BYAO host. Based on the consideration of the effective ionic radii with different coordination numbers [34], the co-doped rare-earth ion Nd^3^^+^ is proposed to occupy the Ba^2^^+^/Y^3^^+^ site in the main lattice of the BAYO.

Since the chemical valence of Eu and Nd can greatly affect the luminescence performance, it is crucial to determine the valence states of the Eu and Nd ions in the BYAO host. The valence states of Eu and Nd were studied using XPS. As shown in Figure 3, the XPS measurements exhibit binding energies corresponding to Ba 3d, Y 3d, Al 2p, O 1s, Eu 3d, and Nd 3d [35,36,37]. Furthermore, as shown in Figure 3e, the high-resolution (HR) XPS depicts the binding energy of the Eu 3d (1125.09 eV) signal, which is consistent with that of Eu^2^^+^ 3d_5/2_ [38,39,40]. The binding energy of Nd 3d peaks at 977.2 eV (Nd 3d_5/2_) and 1000.6 eV (Nd 3d_3/2_) as can be seen in Figure 3f [41,42,43,44,45,46]. Figure 4a and Figure 5a show scanning electron microscopy (SEM) pictures of the BYAO:0.03Eu^2^^+^ and BYAO:0.03Eu^2^^+^, 0.02Nd^3^^+^. The irregularities in the shape, size, and pores of these samples may be related to the irregular mass flow and inhomogeneous temperature distribution of the samples during combustion [47]. The EDS of the BYAO:0.03Eu^2^^+^ and BYAO:0.03Eu^2^^+^, 0.02Nd^3^^+^ phosphors is displayed in Figure 4b and Figure 5b, and the EDS intensity signals of Ba, Y, Al, O, Eu, and Nd are in good agreement with the chemical composition of the BYAO:0.03Eu^2^^+^ and BYAO:0.03Eu^2^^+^, 0.02Nd^3^^+^ phosphors. Moreover, from the mapping analysis in Figure 4c and Figure 5c, it is clearly observed that Eu and Nd are uniformly distributed over the BYAO host.

### 3.2. Photoluminescence and Afterglow Properties

Figure 6a,b display the UV–vis absorption spectra of the BYAO:0.03Eu^2^^+^ and BYAO:0.03Eu^2^^+^, 0.02Nd^3^^+^. The bandgap (*E*_g_) can be calculated on the basis of the following equation: (1)$${\left(\alpha h\nu \right)}^{2}=A\left(h\nu -E_{g}\right)$$
*hν* is the photon energy, α is the absorption coefficient, *E_g_* is band gap energy (eV), and *A* is a constant [40,48,49]. The insets show the experimental values (*E*_g_), which are approximately 4.92 eV and 5.26 eV. The optical properties of the BYAO:Eu^2^^+^ and BYAO:Eu^2^^+^, Nd^3^^+^ phosphors were analyzed by analyzing PLE and PL spectra. As shown in Figure 6c, the PLE spectrum of BYAO:Eu^2^^+^, Nd^3^^+^ shows a broad absorption band centered at 365 nm, which is mainly due to the 4f^7^→4f^6^5d transition of Eu^2^^+^ ions [50,51,52,53]. Under the 365 nm UV light excitation, the PL spectrum of the BYAO:0.03Eu^2^^+^ (~497 nm) and BYAO:0.03Eu^2^^+^, 0.02Nd^3^^+^ (~492 nm) shows the 4f^6^5d→4f^7^ transitions of Eu^2^^+^ ions [53,54,55]. Because of the relatively small crystal-field splitting energy for Eu^2^^+^ ions in the BYAO host crystal, the light emission from the BYAO:Eu^2^^+^ and BYAO:Eu^2^^+^, Nd^3^^+^ phosphors is considerably shorter wavelength, peaking at ~500 nm, than the various Eu^2^^+^-doped phosphors [53]. The intensity of the Eu^2^^+^ single-doped sample is higher than that of the Eu^2^^+^, Nd^3^^+^ co-doped samples, and the emission spectrum is slightly blue-shifted after co-doping, indicating that there may be energy storage in the co-doped samples during this process [13]. Furthermore, in the Eu^2^^+^, Nd^3^^+^ co-doped samples, no emission characteristics of Nd^3^^+^ ions were observed, suggesting that Nd^3^^+^ is not acting as a luminescence center but may play the role of a trapping center. To investigate the doping concentration effect on the optical properties of the samples, BYAO:xEu^2^^+^ (x = 0, 0.02, 0.03, 0.04, 0.05, 0.08) phosphors were synthesized. In Figure 7a, under the excitation of 365 nm, the PL spectra of the phosphors doped with different concentrations of Eu^2^^+^ show that the optimum concentration is 0.03Eu^2^^+^. The luminescence intensity is an increase of 29 times the original intensity. To further investigate the effect of Nd^3^^+^ concentration, the typical experiment was performed for BYAO:0.03Eu^2^^+^, xNd^3^^+^ (x = 0.01–0.05). As seen in Figure 7b, the emission intensity reaches its highest when the doping concentration is 0.02. Figure 7c depicts the coherent infrared energy (CIE) chromaticity coordinate positions of the BYAO:0.03Eu^2^^+^, xNd^3^^+^ (0–0.05) phosphors under the excitation wavelength of 365 nm. With the increased Nd ion concentration, the emission color gradually blue-shifted, and the corresponding CIE color coordinates are changed from (0.151, 0.0.305) to (0.148, 0.216).

In addition, Figure 8a shows the luminescence decay curves of BYAO:0.03Eu^2^^+^ and BYAO:0.03Eu^2^+, 0.02Nd^3^^+^ phosphors (λ_ex_ = 365 nm). The decay curve can be well-fitted with a second-order exponential function [56]: (2)$$I\left(t\right)=I_{0}+A_{1}exp\left(-t/\tau_{1}\right)+A_{2}exp\left(-t/\tau_{2}\right))$$
where *I*_0_ represents the background constant, *A*_1_ and *A*_2_ are constants, and τ_1_ and τ_2_ represent the decay times for the fast and slow exponential components, respectively. The average decay time τ* could be obtained using the following equation: (3)$$\tau^{*}\;=\;\left(A_{1}\cdot \tau_{1}^{2}+A_{2}\cdot \tau_{2}^{2}\right)/\left(A_{1}\cdot \tau_{1}+A_{2}\cdot \tau_{2}\right)$$

The average decay time τ* of the BYAO:0.03Eu^2^^+^ and BYAO:0.03Eu^2^^+^, 0.02Nd^3^^+^ phosphors is 850 ns and 1149 ns, respectively, which provides experimental evidence for the presence of an energy transfer process between Eu^2^^+^ ions and Nd^3^^+^ ions [57]. To further investigate the afterglow properties of phosphors, the afterglow decay curve of the BYAO:0.03Eu^2^^+^, 0.02Nd^3^^+^ samples was measured. As shown in Figure 8b and Figure 9, a bright blue-green afterglow can be observed after exposing the BYAO:0.03Eu^2^^+^, 0.02Nd^3^^+^ phosphor to a 365 UV light for 5 min, and the afterglow time is more than 8 min, which shows the potential application as a nighttime security marker. The afterglow curve can be fitted by the double-exponential Equation (Equation 2). According to the above equation, τ_1_ and τ_2_ are 5.4 s and 51.3 s, respectively. Long afterglow decay occurs in two stages: slow decay and quick decay. These quick decay processes appear first and dominate the intensity. Slow decay processes take place later and result in long-term luminescence behavior [58,59].

The long afterglow is a result of the gradual release of charge carriers trapped in the material, with the afterglow duration and intensity depending on the concentration and depth of the trapping centers. Shallow traps are adversely affected by stable charge carriers, significantly reducing the duration of persistent luminescence. Conversely, charge carriers captured by deep traps are difficult to release at room temperature, also adversely impacting the persistence of luminescence [2]. Therefore, to investigate the afterglow process in detail, the trap information of the BYAO:0.03Eu^2^^+^, xNd^3^^+^ (x = 0.01–0.05) samples was analyzed using TL spectra, as shown in Figure 8c. The samples were exposed to 365 nm UV light pre-irradiation for 5 min at room temperature (30 °C), heating to 500 °C, quickly cooling to 30 °C, and final TL measurement at a heating rate of 1 °C/s. With the increase in the concentration of Nd ions, the high-temperature peak gradually shifts to higher temperatures, which means that the doping of Nd^3^^+^ ions significantly increases the defect levels. As shown in Figure 8d, the TL curve of BYAO:0.03Eu^2^^+^, 0.02Nd^3^^+^ consists of two broad bands with maxima at 94 °C and 176 °C, which correspond to the shallow and deep traps, respectively. The depth of the trap can be estimated by the following equation [60,61,62]: (4)$$E=\frac{T_{m}}{500}$$
where *E* represents the activation energy (depth of the trap), *T*_m_ represents the peak temperature: for the shallow trap, *T*_m_ = 94 °C; for the deep trap, *T*_m_ = 176 °C. Therefore, the depths of the deep and shallow traps in BYAO:0.03Eu^2^^+^, 0.02Nd^3^^+^ are evaluated to be 0.73 eV and 0.9 eV, respectively.

### 3.3. Mechanism for the Afterglow of the BYAO:Eu^2^^+^, Nd^3^+ Phosphor

When Nd^3^^+^ ions replace Ba^2^^+^ ions in the matrix, a positive charge center is generated due to the need to maintain charge balance, that is, a trap is created. Figure 10 shows the possible mechanism for the formation of efficient blue-green afterglow in Eu^2^^+^, Nd^3^^+^ co-doped BYAO. When the UV photon excites a sample, electrons are excited from the ground state to the excited state or conduction band (CB). The excited electrons will transit back to the ground state and recombine with holes to emit light. The electrons in the CB will become free electrons, and after the light irradiation stops, the free electrons in the CB will partly relax to the excited state and subsequently transit back to the ground state and recombine with holes to luminesce. Another part of the free electrons in the CB will be captured by the shallow and deep traps. The electrons that enter the shallow trap slowly escape due to thermal excitation and relax to the CB. Relaxation to the CB of part of the free electrons by the trap then occurs, and they are then captured, and part of the free electron relaxation to the excited state takes place, whereupon they transition back to the ground state and recombine with holes, resulting in afterglow luminescence. Electrons in deep traps also slowly escape and relax into shallow traps due to thermal excitation. The electrons captured in the deep traps do not escape directly into the CB at room temperature, and so the deep trap has a storage function.

### 3.4. Anti-Counterfeiting Application

To further investigate the anti-counterfeiting properties of phosphor, we prepared anti-counterfeiting inks by mixing phosphor with polyacrylic acid and then printed four-leaf clover and snowflake patterns with the screen printing technique. As shown in Figure 11a,b, BYAO/ink is almost invisible under daylight, and clear patterns can be seen under the 365 nm UV excitation. As shown in Figure 11c,d, the pattern printed by BYAO:0.03Eu^2^^+^, 0.02Nd^3^^+^/ink can still be seen clearly after the UV irradiation is turned off. In addition, the afterglow effect can still be seen after the UV excitation has stopped for three minutes. From the above findings, we suggested that the Eu^2^^+^ ions and Nd^3^^+^ ions activating the BYAO ink are anticipated for high-level anti-counterfeiting applications.

## 4. Conclusions

In summary, a series of BYAO:Eu^2^^+^ and BYAO:Eu^2^^+^, Nd^3^^+^ phosphors was prepared via the combustion method at a reaction temperature of 600 °C. The Eu^2^^+^ ions and Nd^3^^+^ ions were successfully doped into the BYAO host, as was verified by XRD, EDS, and XPS analysis. The emission bands of BYAO:Eu^2^^+^ and BYAO:Eu^2^^+^, Nd^3^^+^ are centered at 497 and 492 nm, respectively, which are attributed to the 4f^6^5d→4f^7^ transition of Eu^2^^+^ ions. The fluorescence lifetimes of the BYAO:0.03Eu^2^^+^ and BYAO:0.03Eu^2^^+^, 0.02Nd^3^^+^ are 850 ns and 1149 ns, respectively. In addition, Eu^2^^+^ ions as the emission center and co-doped Nd^3^^+^ ions as the trap center can improve the afterglow performance. The afterglow duration of the BYAO:0.03Eu^2^^+^, 0.02Nd^3^^+^ phosphor can reach 8 min. Finally, a transparent ink was prepared by mixing fluorescent powder with ethanol and polyacrylic acid to demonstrate the potential of fluorescent powder for anti-counterfeiting applications.

## Figures and Tables

**Figure 1 nanomaterials-13-02457-f001:**
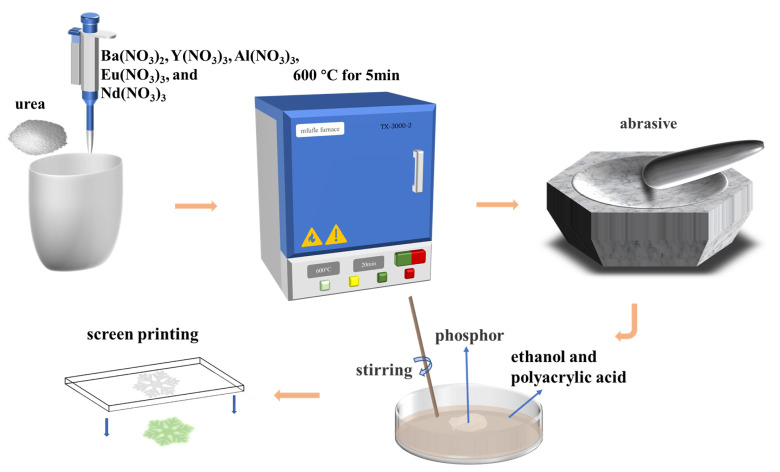
Schematic diagram of ink preparation and screen printing.

**Figure 2 nanomaterials-13-02457-f002:**
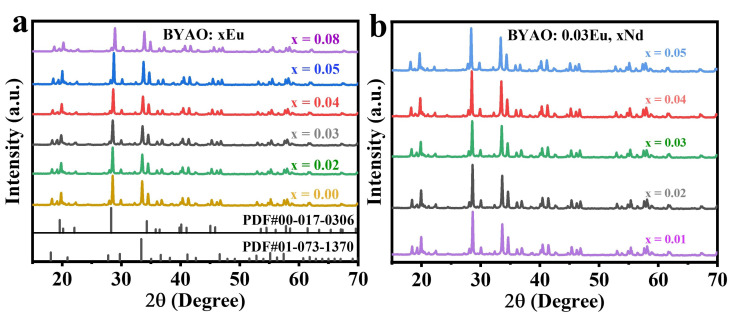
(**a**) The BYAO:xEu^2^+ (x = 0, 0.02, 0.03, 0.04, 0.05, 0.08) XRD patterns and the standard card. (**b**) The XRD patterns of BaYAl_3_O_7_:0·03Eu^2^+, xNd^3^+ (x = 0.01, 0.02, 0.03, 0.04, 0.05) phosphors.

**Figure 3 nanomaterials-13-02457-f003:**
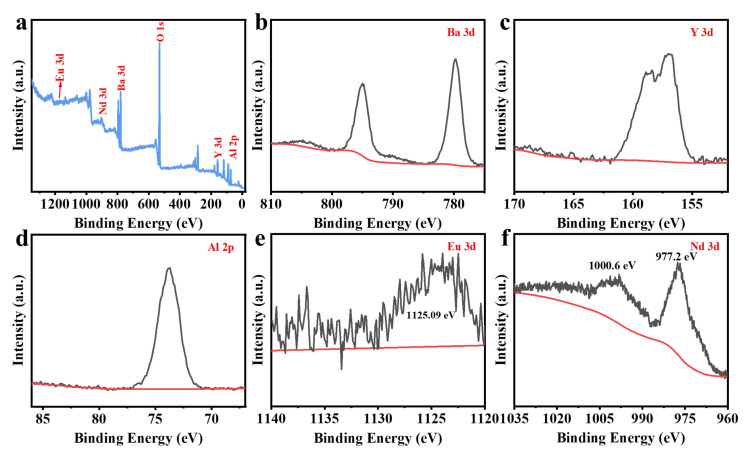
(**a**) The XPS spectra of the BYAO:0·03Eu^2^+, 0·02Nd^3^+. (**b**–**f**) The HR XPS spectra of Ba 3d, Y 3d, Al 2p, Eu 3d, and Nd 3d.

**Figure 4 nanomaterials-13-02457-f004:**
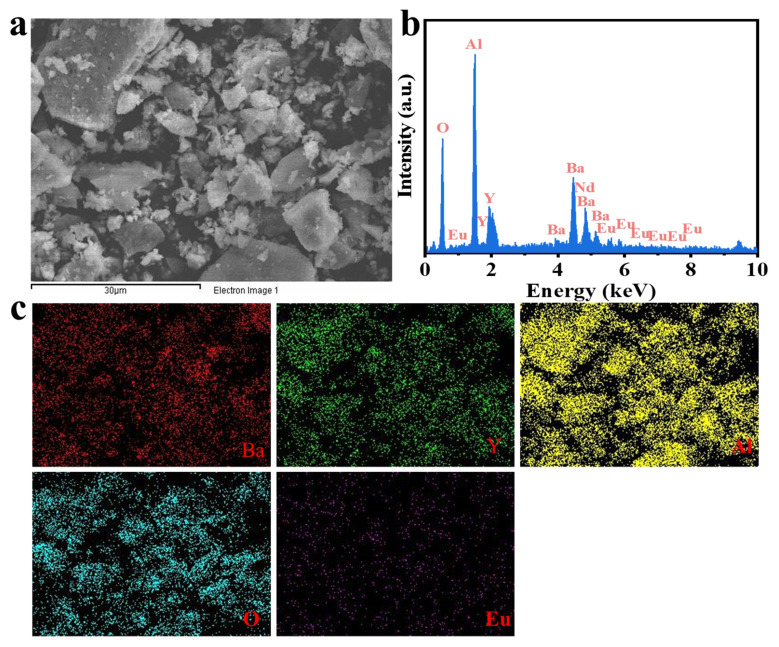
(**a**) The SEM image of the BYAO:0.03Eu^2^+ material. (**b**) The EDS of the BYAO:0.03Eu^2^+ material. (**c**) The elemental distribution of the BYAO:Eu^2^+ material.

**Figure 5 nanomaterials-13-02457-f005:**
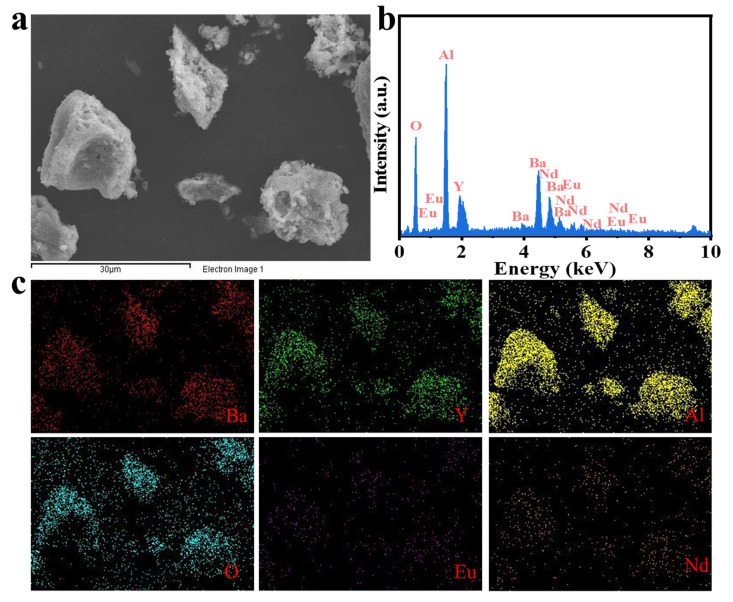
(**a**) The SEM image of the BYAO:0.03Eu^2^+, 0.02Nd^3^+ material. (**b**) The EDS of the BYAO:0.03Eu^2^+, 0.02Nd^3^+ material. (**c**) The elemental distribution of the BYAO:0.03Eu^2^+, 0.02Nd^3^+ material.

**Figure 6 nanomaterials-13-02457-f006:**
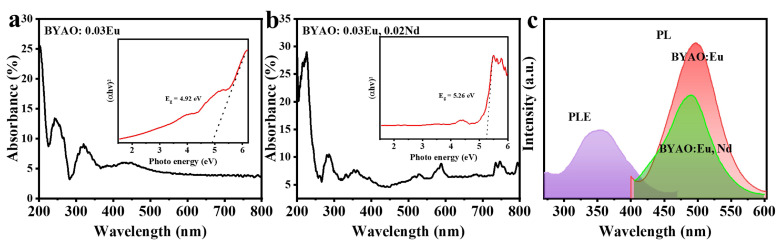
(**a**,**b**) UV–vis absorption spectrum of the BYAO:0.03Eu^2^+ and BYAO:0.03Eu^2^+, 0.02Nd^3^+. The inset shows the theoretical fit of the band gap. (**c**) The emission spectra (λ_ex_ = 365 nm) and excitation spectra (λ_em_ = 496 nm) of the samples.

**Figure 7 nanomaterials-13-02457-f007:**
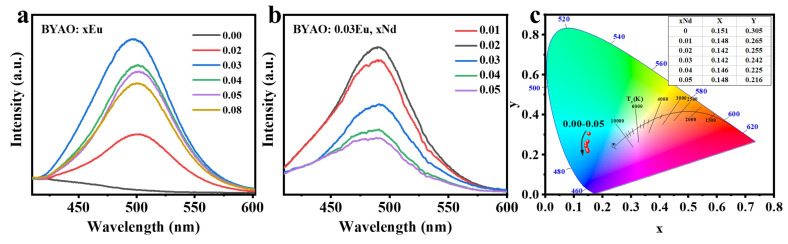
(**a**,**b**) The emission spectra (λ_ex_ = 365 nm) of the BYAO:xEu^2^+ (x = 0, 0.02, 0.03, 0.04, 0.05, 0.08) and BYAO:0.03Eu^2^+, xNd^3^+ (x = 0.01, 0.02, 0.03, 0.04, 0.05). (**c**) CIE chromaticity diagram of the BYAO:0.03Eu^2^+, xNd^3^+ (0–0.05) under the excitation of 365 nm ultraviolet light.

**Figure 8 nanomaterials-13-02457-f008:**
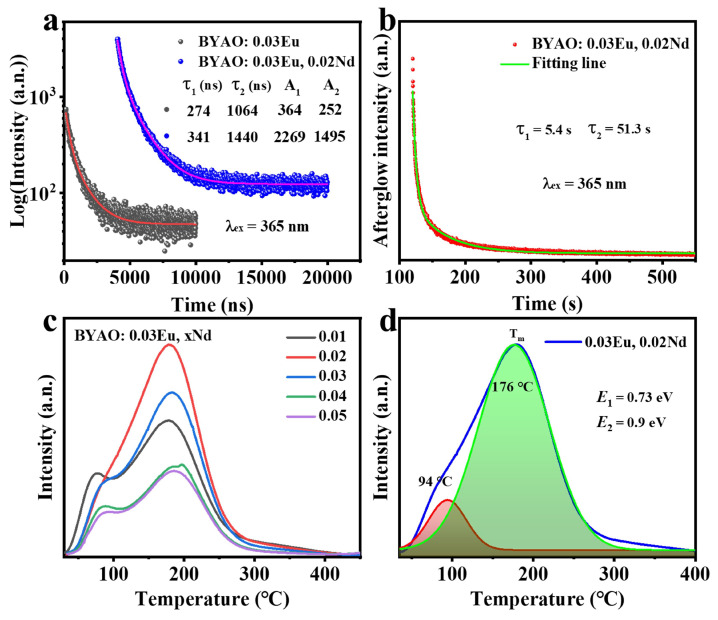
(**a**) The luminescence decay curves of the BYAO:0.03Eu^2^+ and BYAO:0.03Eu^2^+, 0.02Nd^3^+ phosphors. (**b**) Afterglow decay curve of the BYAO:0.03Eu^2^+, 0.02Nd^3^+ phosphor after exposure to a 365 nm UV light for 2 min. (**c**) Concentration-dependent TL curves of the BYAO:0.03Eu^2^+, xNd^3^+ (x = 0.01–0.05) excited for 5 min. (**d**) TL spectrum of the BYAO:0.03Eu^2^+, 0.02Nd^3^+ and its deconvolution into the two Gaussian components at 94 and 176 °C.

**Figure 9 nanomaterials-13-02457-f009:**
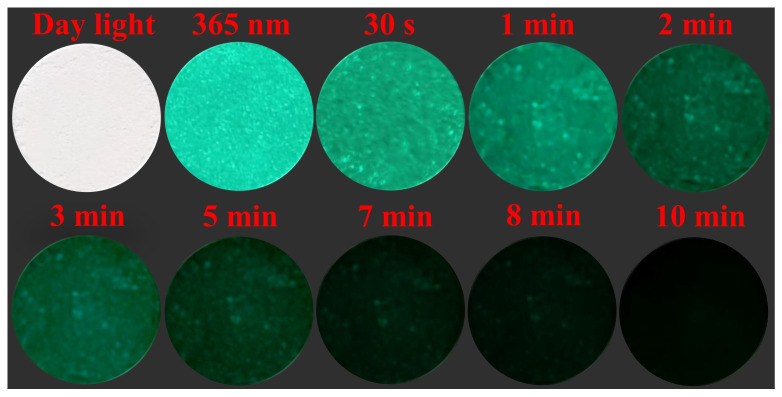
Afterglow photographs of BYAO:0.03Eu^2^+, 0.02Nd^3^+ phosphor after exposure to the 365 nm UV source for 5 min.

**Figure 10 nanomaterials-13-02457-f010:**
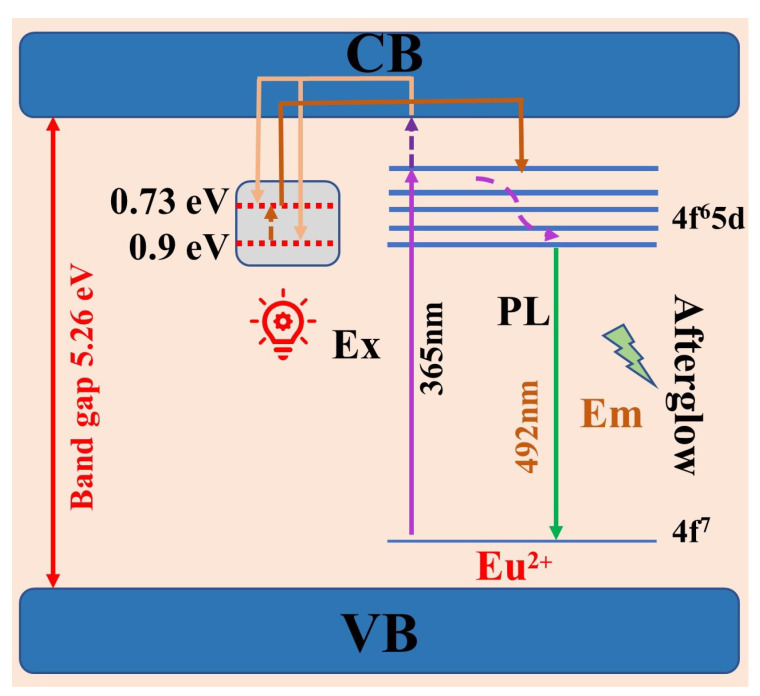
Schematic afterglow mechanism in the BYAO:Eu^2^+, Nd^3^+ phosphor.

**Figure 11 nanomaterials-13-02457-f011:**
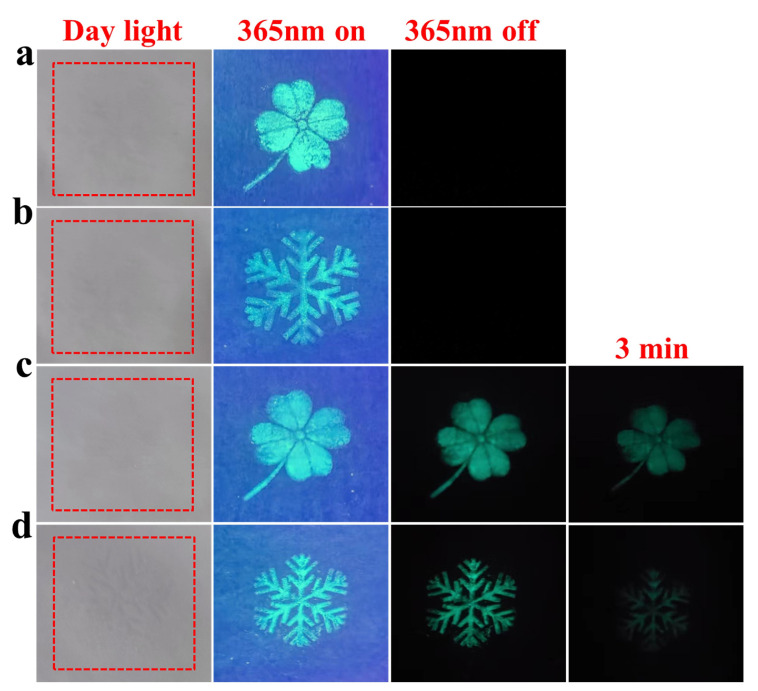
BYAO:0.03Eu^3^+/ink for printing (**a**) four-leaf clover and (**b**) snowflake patterns; BYAO:0.03Eu^2^+, 0.02Nd^3^+/ink for printing (**c**) four-leaf clover and (**d**) snowflake patterns.

## Data Availability

Research data are not shared.

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
