# Peer review of "Design, Synthesis, and Characterization of a Novel Blue-Green Long Afterglow BaYAl3O7:Eu2+, Nd3+ Phosphor and Its Anti-Counterfeiting Application"

_nanomaterials, 2023, doi:10.3390/nano13172457_

Round 1

Reviewer 1 Report

The work "Design, synthesis, and characterization of a novel blue-green long afterglow BaYAl3O7:Eu2+, Nd3+ phosphor and anti-counterfeiting application" is an interesting continuation of the series of works on the preparation and study of the properties of various phosphors and their applications.

The work will be more interesting if the authors heed the following remarks:

1. In the introduction, it is necessary to more fully reflect the advantages of the combustion method chosen for synthesis (10.1134/S0036023618100157).

2. Al2O3 (line 56) or Al(NO3)3 (line 63) was used as raw material?

3. Line 67. To what temperature was the oven heated? It is then clear from the text that 600C, but it must also be indicated on line 67.

4. Are the synthesized particles nanosized, if so, what is their range?

Minor editing of English language required

Reviewer 2 Report

My comments are in attached file.

English could be improved.

Reviewer 3 Report

See attached Comments

See attached Comments

Reviewer 4 Report

The authors well investigated the properties including excitation, emission, and afterglow. The paper should be accepted.

Author Response

We are very grateful for the comments and support provided by the reviewers and wish them much success in their academic research and other fields.

Round 2

Reviewer 3 Report

See Comments_R (PDF file)

Round 3

Reviewer 3 Report

Accept